# Longitudinal Magnetic Resonance Imaging of Cerebral Microbleeds in Multiple Sclerosis Patients

**DOI:** 10.3390/diagnostics10110942

**Published:** 2020-11-12

**Authors:** Karthikeyan Subramanian, David Utriainen, Deepa P. Ramasamy, Sean K. Sethi, Ferdinand Schweser, John Beaver, Jesper Hagemeier, Bianca Weinstock-Guttman, Rajasimhan Rajagovindan, Robert Zivadinov, Ewart Mark Haacke

**Affiliations:** 1Department of Radiology, Wayne State University, Detroit, MI 48201, USA; sethisea@wayne.edu (S.K.S.); nmrimaging@aol.com (E.M.H.); 2Magnetic Resonance Innovations, Inc., Bingham Farms, MI 48025, USA; david@spintechimaging.com; 3SpinTech, Inc., Bingham Farms, MI 48025, USA; 4Buffalo Neuroimaging Analysis Center, Department of Neurology, Jacobs School of Medicine and Biomedical Sciences, University at Buffalo, The State University of New York, Buffalo, NY 14203, USA; dramasamy@bnac.net (D.P.R.); schweser@buffalo.edu (F.S.); jhagemeier@bnac.net (J.H.); rzivadinov@bnac.net (R.Z.); 5Translational Imaging Center at Clinical and Translational Science Institute, University at Buffalo, The State University of New York, Buffalo, NY 14203, USA; 6Translational Imaging, Abbvie Inc., North Chicago, IL 60085, USA; john.beaver@abbvie.com (J.B.); rajasimhan@gmail.com (R.R.); 7Jacobs Multiple Sclerosis Center, Department of Neurology, Jacobs School of Medicine and Biomedical Sciences, University at Buffalo, The State University of New York, Buffalo, NY 14202, USA; BWeinstockGuttman@kaleidahealth.org

**Keywords:** multiple sclerosis, cerebral microbleeds, susceptibility mapping, susceptibility weighted imaging

## Abstract

We hypothesized that cerebral microbleeds (CMBs) in multiple sclerosis (MS) patients will be detected with higher prevalence compared to healthy controls (HC) and that quantitative susceptibility mapping (QSM) will help remove false positives seen in susceptibility weighted imaging (SWI). A cohort of 100 relapsing remitting MS subjects scanned at 3T were used to validate a set of CMB detection guidelines specifically using QSM. A second longitudinal cohort of 112 MS and 25 HCs, also acquired at 3T, was reviewed across two time points. Both cohorts were imaged with SWI and fluid attenuated inversion recovery. Fourteen subjects in the first cohort (14%, 95% CI 8–21%) and twenty-one subjects in the second cohort (18.7%, 95% CI 11–27%) had at least one CMB. The combined information from SWI and QSM allowed us to discern stable CMBs and new CMBs from potential mimics and evaluate changes over time. The longitudinal results demonstrated that longer disease duration increased the chance to develop new CMBs. Higher age was also associated with increased CMB prevalence for MS and HC. We observed that MS subjects developed new CMBs between time points, indicating the need for longitudinal quantitative imaging of CMBs.

## 1. Introduction

Cerebral microbleeds (CMBs) are small foci of chronic blood products in normal or near normal brain tissue [1]. They represent a radiologic construct which correlates with histopathology [2,3], specifically hemosiderin deposition from past hemorrhages that manifest as focal hypo-intense lesions on T2*-weighted gradient echo (GRE) images. Increased prevalence of CMBs has been demonstrated in neurologically healthy elderly populations [4,5], mild cognitive impairment, dementia and Alzheimer’s disease (AD) [6], vascular dementia [7,8], patients receiving anti-thrombotic medications [9,10], chemotherapy [11], hypertension [12], and as a risk factor for developing stroke [13]. CMBs have also been associated with decreased cognitive performance in both traumatic brain injury (TBI) [14] and stroke patients [15].

Sensitivity and specificity to detect CMBs are influenced by the spatial resolution and slice thickness relative to the true size of CMBs. Susceptibility Weighted Imaging (SWI) is more sensitive compared to 2D T2* GRE imaging [16,17,18] in detecting CMBs; however, it is limited in its ability to distinguish the magnetic state (diamagnetic vs. paramagnetic properties) of an object. The use of phase information and iterative Susceptibility Weighted Imaging and Mapping (iSWIM) [19,20] (a form of Quantitative Susceptibility Mapping (QSM) [21]) can overcome this limitation and, therefore, the signal source type (calcium or iron) can be differentiated and quantified [21].

Multiple Sclerosis (MS) is an immune-mediated demyelinating and neuro-degenerative disorder of the central nervous system that is characterized by blood-brain barrier (BBB) disruption, inflammation, lesion formation, and development of central nervous system atrophy [22]. MS is a progressive disease with increased prevalence of cardiovascular diseases that are also risk factors for CMBs [23]. A recent study by Zivadinov et al. [24] showed that more patients with MS than HCs had CMBs (19.8% vs. 7.4%, respectively; *p* = 0.01) in a group at least 50 years of age or older. That study demonstrated a trend toward greater presence of CMBs in patients with MS (*p* = 0.016) and patients with clinically isolated syndrome (CIS) who were younger than 50 years (*p* = 0.039) compared with HCs. Regression analysis indicated that CMB counts were positively associated with physical disability and that cognitive disability was associated with CMBs in MS. They further demonstrated that cardiovascular risk factors were a key determinant in the development of CMBs.

In this study, we used the microbleed guidelines from Greenberg et al. [1] along with the information provided by Haacke et al. [21] incorporating the use of iSWIM to detect and differentiate CMBs from other mimics. In doing so, our goal was to create an augmented set of CMB guidelines alongside our CMB findings in this longitudinal MS cohort. We initially studied 100 Relapsing Remitting Multiple Sclerosis (RRMS) cases from our neuroimaging database [25] to evaluate CMB detection using these guidelines. We then used these guidelines to review a separate cohort of 112 MS patients [24] with scans from two time points to confirm and extend the previous findings. These 112 patients were first included in the clinically oriented paper by Zivadinov et al. [24] in order to better understand the prevalence and factors which contribute to CMB in MS and healthy controls. The intent of this work was to include QSM in the CMB detection process and to explore the development of new CMBs longitudinally (something not done in the previous study [24]). We hypothesize that the addition of iSWIM data to CMB analysis will yield higher sensitivity and specificity compared with SWI data alone [26].

## 2. Materials and Methods

Two cohorts are considered in this study. A group of 100 RRMS patients imaged on a Siemens 3T scanner was used to establish an improved set of CMB detection guidelines and another group of 112 MS patients from a General Electric 3T scanner was evaluated for longitudinal changes in the number of microbleeds. Data usage was approved by the local Institutional Review Boards and informed consent was obtained from all subjects.

Cohort 1: A set of 100 RRMS patients were randomly selected from an imaging database [25]. A retrospective analysis of the data was used to establish an improved set of the conventional CMB guidelines [1] by including the iSWIM data. This group included 72 female and 28 male patients, with a mean age of 47.5 ± 10 years, and disease duration of 11.9 ± 8 years. The inclusion criteria for this sub-study of CMBs from the 100 RRMS patients were: (a) having a baseline SWI data acquisition, (b) an age of 18–75 years, and (c) an MR imaging examination within 30 days of physical and/or neurological examination with the standardized study protocol described below. The inclusion criteria for the MS subjects were that the subjects need to have a definite MS condition with no other existing neurological diseases. Exclusion criteria were: (a) pregnancy, (b) evident relapse in the last 30 days prior to screening, (c) clinically significant brain abnormality other than MS diagnosis (i.e., any cerebrovascular disease), or (d) clinically unstable with any history of major illness (i.e., prior known neurological disorders other than MS, alcohol or substance abuse).

Cohort 2: For purposes of determining the longitudinal incidence in CMBs, 25 healthy controls (HC) and 112 MS patients from a prior study [24] (93 RRMS, 15 Secondary Progressive (SP) MS, 2 Primary Progressive (PP) MS, 1 Progressive Relapsing (PR) MS and 1 Radiologically Isolated Syndrome (RIS)) were randomly selected (none of whom had neurological conditions apart from MS). All subjects had been scanned twice with an interval between 1 and 4 years. This group included 76 female and 36 male patients, with a mean age of 45.4 ± 10 years (age at first scan) and disease duration of 12.7 ± 9 years. The Expanded Disability Status Scale (EDSS) was 2.9 ± 1.7 at baseline and 3.1 ± 1.7 at follow-up.

The MR data for cohort 1 was acquired on a Siemens 3T TRIO Scanner (Erlangen, Germany) using a twelve-channel head coil and the MRI for cohort 2 was acquired on a General Electric (GE) 3T Signa Excite HD 12.0 scanner (Milwaukee, WI, USA) using an eight-channel head coil. The protocol included: pre/post-contrast T1 weighted imaging (T1WI), T2 weighted imaging (T2WI), T2 fluid attenuated inversion recovery (FLAIR), and susceptibility weighted imaging (SWI) (including the original phase data). The imaging parameters for both data sets are shown in Table 1 and Table 2. For cohort 2 the SWI data were interpolated to 0.5 mm × 0.5 mm in-plane resolution. All images were post-processed using SPIN software (SpinTech Inc., Bingham Farms, MI, USA). In brief, the QSM data were processed via iSWIM as follows: the brain extraction tool (BET) was used (threshold = 0.2, erode = 4 and island = 2000) followed by 3D phase unwrapping with sorting by reliability, following a non-continuous path (3DSRNCP). Then the sophisticated harmonic artifact reduction (SHARP) filter was used to remove unwanted background fields (threshold = 0.05 and deconvolution kernel size = 6) and, finally, a truncated k-space division (TKD) based iterative inverse filtering technique (threshold = 0.1) [19] was used (iteration threshold = 0.1 and number of iterations = 4) to reconstruct the susceptibility maps [20]. This form of QSM is referred to as iterative susceptibility weighted imaging and mapping (iSWIM).

CMB Detection Guidelines: The following guidelines were used to identify CMBs while avoiding potential mimics. Specifically, the CMBs should: (1) be round or ovoid (rather than linear) in-plane and through plane; (2) be dark on T2*W imaging; (3) demonstrate a dipole effect in SWI phase images; (4) appear bright in iSWIM data (representing a paramagnetic substance); (5) be isolated, as can be verified using minimum/maximum intensity projections (mIP/MIP) for SWI/iSWIM, respectively; (6) be at least half surrounded by brain parenchyma or CSF and not air; (7) not be connected to venous structures and (8) be distinct from other potential CMB mimics. For rule 7, a positive result for vascular damage (TBI) may be interpreted as intravascular medullary vein thrombosis especially if there is clear venous connectivity to this region. For rule 8, mimics include mineralization other than iron with opposite susceptibility properties (diamagnetic, as is the case for calcifications), flow voids, cusp artifacts, and iron appearing within macrophages in MS lesions. We consider CMBs to have a diameter less than or equal to 5 mm. The location of the CMB was then compared to its appearance in T2WI, T2 FLAIR and pre and post contrast T1WI images.

Two image processing raters with two and five years of MRI signal processing experience were included in an inter-rater reliability test for both CMB detection and location using the MARS (Microbleed Anatomical Rating Scale) [27] scale from cohort 1. For the longitudinal study, two raters with two and four years of experience, were blinded, and the same above-mentioned guidelines were used to detect the CMBs. The raters were later unblinded and their results compared against each other. Raters were considered reliable if the intra-class correlation (ICC) statistic 2 was greater than 0.9 for CMB number, iron content and volume size. Reliability testing for location was done with a kappa statistic. All statistical analysis was done using SPSS software (Version 20 (IBM Corp, Armonk, NY, USA)). This included the ICC statistic as well as bootstrapping of the CMB frequency treating the information obtained by retrospective analysis of the group of 100 RRMS with 5000 iterations.

## 3. Results

### 3.1. Cohort 1

A total of 18 CMBs were found in 14 subjects, with four subjects having two CMBs each. All 18 CMBs were visualized in the original unprocessed long echo magnitude images (T2*), the phase images, the SWI composite images and iSWIM. None of the CMBs in cohort 1 were co-localized with existing MS lesions such as acute breakdown of the BBB as observed by parenchymal contrast agent leakage in T1WI, T1WI hypo-intensities, or hyper-intensities in T2 FLAIR. This corresponds to a 14% (14/100) overall prevalence rate. The bootstrapping and resampling yielded a 95% confidence interval of 8–21% for CMB prevalence. Two other subjects had signals which mimicked a bleed which were actually a part of a vessel and not a CMB. This was recognized with the use of mIP/MIP on SWI/iSWIM, respectively. Further, iSWIM aided in differentiating 5 different calcium deposits (3 as clear microbleed mimics and 2 in the pineal gland area as potential microbleed mimics) from CMBs in 5 different subjects (5% of the population). Also, iSWIM helped to differentiate thrombus from CMBs by using their connectivity to vessels and its high susceptibility value. Thus, we emphasize that the addition of QSM to the CMB detection guidelines helps to distinguish CMBs from CMB mimics. Some example mimics are shown in Figure 1.

### 3.2. Cohort 2

In the longitudinal cohort, 21/112 MS patients had CMBs in either of the two scans of which 16 of 93 were RRMS (17.2%), 4 of 15 were SPMS (26.6%) and 1 of 2 was PPMS (50%). A total of 20 CMBs in 16 subjects were found in the first scan. A total of 28 CMBs in 20 subjects were found in scan 2. The raters detected all the CMBs in the blinded study (cohort 2). Each rater correctly listed their location with a kappa statistic value of 0.95 as listed in Table 3. All the CMBs were visualized in the original long echo magnitude images (T2*), the phase images, the SWI composite images and iSWIM. This corresponds to an 18.7% overall prevalence rate which lies within the confidence interval from the first set. Bootstrapping and resampling from the second set yielded a 95% confidence interval of 11–27% for CMB prevalence. A combined bootstrapping and resampling was done on both the sets together (100 + 112 = 212 individuals) which yielded a 95% confidence interval of 12–22%.

In the blind assessment, the raters were misled by two mimics when the iSWIM images were absent from the analysis. These mimics were later confirmed as calcium deposits which appeared dark in iSWIM (diamagnetic). The blind assessment showed that SWI and iSWIM were the best modalities to detect all the CMBs without any false positives when used together.

### 3.3. Longitudinal CMB Analysis

In Cohort 2, five patients without CMBs during scan one developed one CMB each at their follow-up. One of these five patients who developed new bleeds had the bleeding inside an iron laden multiple sclerosis lesion [28]. One female patient aged 46 years had a bleed in the first scan, but it disappeared in the second scan. Three of the 16 subjects (18.7%) who had a bleed at scan 1 developed new bleeds at their follow-up scan. A male patient aged 50 years had three CMBs at baseline, and two more CMBs were observed at the follow up. Another male patient aged 60 years who had one CMB in the first scan developed a new one by the second scan (Figure 2). A female patient aged 56 years had one CMB at baseline and developed a second CMB at follow-up. Another 44-year-old female subject at baseline scan had a bleed which increased in diameter and intensity by the second scan (Figure 2). A total of 8 out of the 112 MS patients (7.1%) developed new CMBs between the two scans. One normal control female subject aged 52 years had a cerebellar CMB at the follow-up scan.

### 3.4. CMB vs. Subject Demographics

A typical age bias is observed in CMB prevalence, with CMBs increasing in number in the healthy population after the age of 50 years [4]. Age was found to be a factor associated with higher CMB prevalence in the MS population as well. Thirteen percent (8/61) of patients aged less than 50 years had CMBs at the follow-up and 25.5% (13/51) of patients aged 50 years or older showed CMBs at the follow-up. Table 4 shows an increased population of subjects with CMBs after the age of 50 years. Also, when a t-test statistic was done, a difference in the average age of people with CMBs was seen when compared to those without CMBs (*p* < 0.05).

In cohort 1, 15% (11/72) of females compared to 11% (3/28) of males had CMBs. In the longitudinal study of 112 patients, 14% (11/76) of females and 27.7% (10/36) of males showed CMBs. A t-test result (*p* = 0.40) between the sex of people with CMBs and those without CMBs displayed no difference (*p* > 0.05). The subjects who developed new microbleeds had a maximum EDSS change of 0.5 between the two scans. The average change of EDSS score in the non-CMB population was 0.2. In cohort 2, only one RRMS among the 21 MS subjects with CMB had a relapse in the last year. None of the other MS subjects with CMBs had any relapses.

## 4. Discussion

### 4.1. CMB Detection and Guidelines

Uncertainties in detecting CMBs stem from false positives or false negatives. Calcium deposition appears as a dark region on T2*W scans, similar to CMBs, and are commonly found in the basal ganglia, choroid plexus, pineal gland, falx tentorium, and lobar locations. It can be difficult to differentiate these calcium depositions from CMBs. In this case, iSWIM correctly identified the dark signal as calcium (diamagnetic) while iron appears bright (paramagnetic). iSWIM aided in differentiating these calcium deposits in 5 different subjects in cohort 1 (5%). A second potential mimic leading to a potential false positive is iron deposition in MS lesions, which will appear as dark rings or in some cases small, focal areas of reduced signal intensity on SWI images or as bright areas in iSWIM images. Unless the change in putative iron content of these lesions is greater than surrounding white matter tissue (>50 ppb), it would be challenging to classify these as CMBs. Rather, they may be due to demyelination or iron sequestered by macrophages [29]. A third potential mimic is pial veins caught in cross-section in cortical sulci. These can be distinguished from sulcal CMBs by their linear appearance over contiguous slices or projections. Often veins at the edge of the brain might be misinterpreted as CMBs if a 3D view or minimum (maximum) intensity projection is not used for SWI (iSWIM). The images should be projected into a multi-planar reconstruction to view the suspected CMB in relation to the surrounding tissue and vascular structures from the standard viewing planes. False negatives or simply missing CMBs can occur due to signal loss near air/tissue interfaces obscuring CMBs or hindering a correct interpretation.

In the future, if multiple echo data were available, the image quality for iSWIM data would improve for larger hemorrhages with significant amounts of blood since aliasing and dephasing would be considerably reduced if short echoes were also used and not just TE = 20 ms, for example. Other sources of signal loss due to bleeding exist such as subarachnoid hemorrhage, in and around tumors, cavernous malformations, and telangiectasias. However, these tend to be diffuse and/or greater than 5mm in diameter and are not a confound for CMB detection [30]. Finally, iSWIM offers the potential to see small increases in iron content representing continued bleeding that might not manifest visibly in SWI.

### 4.2. CMB Prevalence in MS and Other Diseases

The estimated prevalence of 14% and 18.7% (95% CI: 8–24% and 11–27% respectively) found in this present MS cohort study is higher than the 6–8% prevalence in neurologically-normal volunteers reported in prior studies [4,31,32]. Eight out of the 112 MS subjects (7.1%) in cohort 2 developed new bleeds and only one out of the 25 healthy controls (4%) developed a new bleed between the two time points. Three of the 16 MS subjects (18.7%) who had a bleed at scan 1 developed new bleeds at their follow-up scan. This indicates that MS subjects have a higher likelihood to develop new CMBs over time. There has long been thought to be some thrombotic involvement in the veins in MS patients. Although clearly not true for all cases, this could be a manifestation of a breakdown of the BBB due to increases in vascular permeability [33]. Also, the risk of developing CMBs is increased by a decline in vascular health. Our study shows that bleeds in MS subjects may grow in size over time (Figure 2). A previous study has shown that MS patients have an increased prevalence of CMBs [24]. Our result from both cohorts not only confirms this but also shows that they change over time.

From a longitudinal imaging perspective, these results suggest that when evaluating MS patients, it is appropriate to use both SWI and QSM to evaluate the presence of CMBs. Due to their small size, high resolution SWI is best suited to detect CMBs on the order of one millimeter [34,35]. A limitation of SWI is its inability to differentiate the source of the T2* signal loss (i.e., calcification versus CMB), therefore, an experienced rater should use QSM to help make this determination.

Development of new CMBs has been observed in other diseases as well. In a cohort of 254 memory clinic patients (including 74 patients with Alzheimer’s disease) who underwent repeated MRI after an average period of two years, one or more new CMBs were observed in 12% of the subjects [36]. In our study, 13% (8/61) of patients aged less than 50 years had CMBs at the follow-up and 25.5% (13/51) of patients aged 50 years or older showed CMBs at the follow-up. This shows an increased prevalence of CMBs in people above 50 years which is in accordance with prior reports. Another study showed that in 21 patients with ischemic stroke, half of the them developed new CMBs after a mean interval of 5.5 years [37]. CMBs are also strongly associated with the presence of intracranial hemorrhage [10]. These numbers are in agreement with a number of other studies in other neurovascular diseases [38,39]. From our own data’s perspective, the combined bootstrapping and resampling result on both the sets combined together (212 subjects) yielded a 95% confidence interval of 12 to 22%. This is comparable to the prevalence of CMBs in mild cognitive impairment of 20% and AD of 18%. On the other hand, vascular dementia rates can go up to 65% [38]. In a study including individuals over 80 years of age, over 38% showed evidence of CMBs [40]. In community-dwelling normal elderly subjects, the prevalence of CMBs is cited to be between 11–23%. Finally, on the opposite end of the spectrum, the presence of CMBs in healthy controls less than 45 years of age has been reported to be only 3% [41]. In our study, none of the HCs under 50 years of age developed a CMB, but one subject developed a CMB who was 52 years old. Overall, these results show that age seems to be a factor in CMB development even in healthy individuals. However, in our study, MS subjects under the age of 50 years had a significantly higher CMB prevalence rate when compared to healthy controls of the same age (13% vs. 0%).

Although a previous study [24] showed a correlation between increased CMB count and clinical outcomes, no correlation was observed between EDSS and CMB prevalence in our study. This may be because the previous study included a large population of SPMS subjects who tended to be older, with longer disease duration and higher EDSS scores. Also, the total populations of these two studies were different. Our study included 93 RRMS, 15 SPMS and 2 PPMS patients, while the other study had 266 RRMS, 138 SPMS and 41 PPMS.

There were several limitations to this study. First, there was a lack of medical history for the subjects such as their medications or if they had any history of cardiovascular diseases which are associated with CMBs. Second, our study was heavily skewed to RRMS and all but one of these had no relapses, so no determination of microbleed prevalence in relapse patients was possible. Third, although we did not have data on cardiovascular risk factors associated with the location of the CMBs, we have reported the location of the CMBs. Fourth, since our study population was dominated by RRMS subjects, further studies regarding the development of new microbleeds longitudinally in SPMS and PPMS subjects will be required to answer this question. Lastly, cohort 2 used a lower resolution than cohort 1. However, we know that the difference in TR (30 ms vs. 40 ms), TE (21 ms vs. 22 ms) and FA (15° vs. 12°) between cohort 1 and 2, respectively, does not cause significant changes to the signal behavior that would affect the CMB detection. The higher resolution of SWI in cohort 1 could, in principle, have found more CMBs, however, as it turns out the second cohort still showed more microbleeds both initially and new CMBs longitudinally even with the lower resolution. Also, even though cohort 2 was imaged with a slightly thicker slice, the mIP/MIP of the SWI/QSM still were effective in distinguishing CMBs from the calcification mimics and mimics associated with vascular structures. The resolution of 0.5 × 1 × 2 mm^3^ has been a standard for nearly twenty years and still provides excellent SWI and QSM results, although higher resolution can improve on QSM image quality and small CMB detection.

## 5. Conclusions

In this work, we used iSWIM as part of the guidelines for CMB detection since susceptibility mapping does not suffer from the geometry dependence of the magnetic fields seen in the phase images. These susceptibility mapping methods offer the ability to quantify the iron content in the CMBs and to rule out calcifications as false positives. There was no correlation between the presence of CMBs and sex or EDSS score for the longitudinal cohort. Our findings showed that MS patients have a high prevalence of CMBs when compared to HCs. Further, the finding that the number of CMBs increased in the MS group between time points may indicate that MS patients, especially those who already have CMBs, may be at a higher risk to develop more CMBs. Age was seen to be another factor for CMB development. MS subjects over the age of 50 years of age had a higher chance of developing CMBs when compared to the ones less than 50 years of age. The prevalence rate in the MS group under age 50 years was higher than that in the HCs of the same age range. Hence, CMBs may serve as a biomarker for neurovascular disease in MS patients, especially in the age group less than 50 years.

## Figures and Tables

**Figure 1 diagnostics-10-00942-f001:**
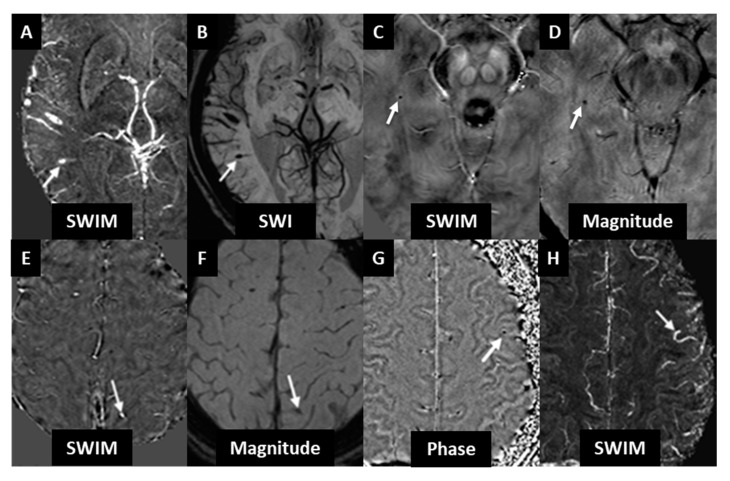
CMB mimics can be differentiated from true CMBs by reviewing their appearance in all SWI modalities simultaneously. (**A**,**B**) Vascular damage indicated by the white arrow can be differentiated by its connectivity to surrounding veins. The SWI minimum intensity projection (over 5 slices) shows a connectivity of the bleed to a draining vein suggesting an intraluminal source for the CMB possibly a thrombus in this case. (**C**,**D**) Magnitude images cannot differentiate diamagnetism from paramagnetism (**D**) as indicated by the white arrow. The same region when viewed in the iSWIM reconstruction appears hypointense (**C**) indicating that the CMB candidate is diamagnetic and that the source is likely a calcification and not a CMB. (**E**,**F**) Potential CMB candidates may be vascular structures as their orientation to the slice and the magnetic field can create phase effects similar to those of a CMB. In the iSWIM reconstruction (**E**) the structure as indicated by the white arrow is shown to have a high paramagnetic susceptibility, however, in the second echo magnitude (**F**) the center of the object lacks clear hypo-intensity indicating it is likely a fast flowing vessel. (**G**,**H**) Veins which are perpendicular to the slice, such as the vein labeled with the white arrow, can appear as round hypo-intensities and may even show a dipole effect in phase (**G**) if it is near a confluence of two veins. In the iSWIM maximum intensity projection (**H**) clear connectivity to a pial vein can be observed and the signal intensity in phase is seen on the QSM result to be from a continuous vessel.

**Figure 2 diagnostics-10-00942-f002:**
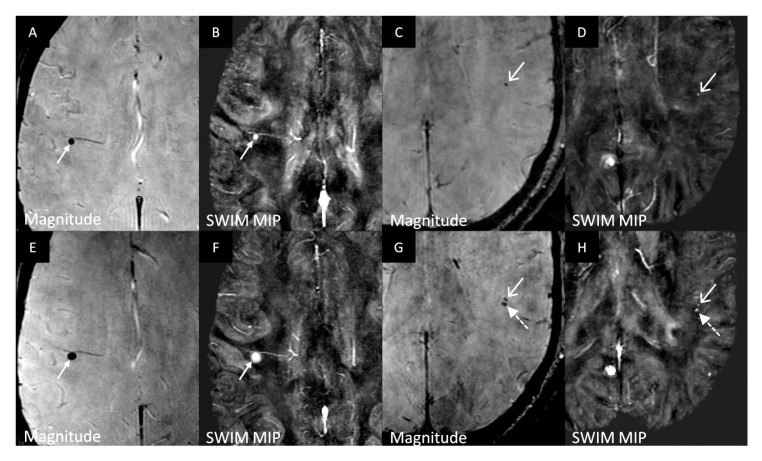
(**A**,**B**) represent the magnitude and iSWIM MIP (over 5 slices) images, respectively, for the first scan while (**E**,**F**) represent the matching images for the follow-up scan, respectively. The CMB is labeled with the white arrow and appears hypo-intense in magnitude and hyper-intense in the iSWIM MIP, indicating it is paramagnetic. The second scan images (**E**,**F**) show an increase in size and intensity of the bleed which indicates that there was further bleeding over time. (**C**,**D**) represent the magnitude and iSWIM MIP (over 5 slices) images for the first scan of a different subject while (**G**,**H**) represent the matching images for the follow-up scan. The white arrows point to a bleed which appears hypo-intense in magnitude and hyper-intense in the iSWIM MIP at both time points. The dotted arrows in the second scan (**G**,**H**) depict a new bleed that developed after the first scan.

**Table 1 diagnostics-10-00942-t001:** Imaging parameters for the sequences used for cohort 1.

Sequence	T2WI	T2 FLAIR	3D T1WI	2D T1WI	SWI	SWI
Orientation	Axial	Sagittal	Axial	Sagittal	Axial	Axial
TR (ms)	7080	6000	1750	306	30	30
TE (ms)	77	396	2.93	9.4	7/21	6/21
FA (degrees)	120	120	9	75	15	15
FOV (mm^2^)	256 × 192	256 × 256	256 × 256	229 × 229	224 × 168	256 × 192
Matrix (N_x_ × N_y_)	512 × 192	258 × 256	512 × 256	384 × 326	448 × 336	512 × 384
Resn (mm^3^)	0.5 × 0.5 × 2	0.5 × 0.5 × 1	0.5 × 0.5 × 1	0.6 × 0.6 × 5	0.5 × 0.5 × 1	0.5 × 0.5 × 1.5
N_z_	100	160	192	20	128	120

T2WI: 2D T2 weighted imaging; T2 FLAIR: 3D T2 Fluid attenuated inversion recovery; T1WI: 2D T1 weighted imaging; SWI: Susceptibility Weighted Imaging; TR: repetition time; TE: echo time; FA: flip angle; FOV: Field-of-view; Resn: resolution; and Nz: number of slices. Some T1WI pre- and post-contrast imaging were collected with turbo spin echo (TSE) in the sagittal orientation with a slice thickness of 5mm compared to the 3D T1-MPRAGE with an inversion time (TI) of 900ms and slice thickness of 1 mm. The TR quoted for T1WI represents the repeat time between inversion pulses. SWI was collected with either 1 mm or 1.5 mm slice thickness.

**Table 2 diagnostics-10-00942-t002:** Imaging parameters for the sequences used for cohort 2.

Sequence	T2WI	T2 FLAIR	3D T1WI	SWI
Orientation	Axial	Axial	Axial	Axial
TR (ms)	5300	8500	5.9	40
TE (ms)	98	120	2.8	22
FA (degrees)	90	90	10	12
FOV (mm^2^)	256 × 192	256 × 192	256 × 192	256 × 192
Matrix (N_x_ × N_y_)	256 × 192	256 × 192	256 × 192	512 × 192
Resn (mm^3^)	1 × 1 × 3	1 × 1 × 3	1 × 1 × 1	0.5 × 1 × 2
N_z_	64	64	184	64

T2WI: 2D T2 weighted imaging; T2 FLAIR: 2D T2 FLuid Attenuated Inversion Recovery; T1WI: 3D T1 weighted imaging; SWI: Susceptibility Weighted Imaging; TR: Repetition time; TE: Echo time; FA: Flip angle; FOV: Field-of-view; Resn: Resolution; and Nz: Number of slices. The SWI data was interpolated to 0.5 mm × 0.5 mm in-plane when creating the final images. The 3D high-spatial-resolution T1-weighted fast spoiled gradient-echo sequence was performed with a magnetization-prepared inversion-recovery pulse with an inversion time (TI) of 900 ms. The TR quoted for the T1WI represents the TR between individual RF pulses during data collection.

**Table 3 diagnostics-10-00942-t003:** CMB distribution in brain regions according to the Microbleed Anatomic Rating Scale (MARS) confirmed by both raters on 112 subjects (cohort 2). A total of 29 microbleeds (from both time point scans) have had their locations categorized in this table.

Brain Regions	Number of CMBs in MS Subjects
Brainstem	2
Cerebellum	3
Frontal	7
Occipital	3
Parietal	6
Temporal	4
Basal ganglia	1
Deep periventricular white matter	2
Insula	0
Thalamus	1
Internal capsule	0
External capsule	0

The number of CMBs also includes the one CMB that disappeared in the second time point scan.

**Table 4 diagnostics-10-00942-t004:** CMB prevalence in patients with MS (cohort 2).

Parameter	Total Patients with MS		Patients > 50 years		Patients < 50 years	
	Scan 1	Scan 2	Scan 1	Scan 2	Scan 1	Scan 2
Patients with CMB	16	20	9	13	7	7
Demographics:						
Mean age (years)	49.3	52.3	56.6	57.7	44.9	42.9
Disease duration (years)	14.8	17.8	18.2	19.9	12.9	15
Mean EDSS	3.7	3.7	3.7	4.3	3.1	2.9
CMB frequency:						
0 CMB	5	1 *	1	0	4	1 *
1 CMB	13	15	7	9	6	6
2 CMB	2	4	1	3	1	1
≥3 CMB	1	1	1	1	0	0

There were 3 subjects who were <50 years of age during scan 1 and were >50 years of age during scan 2. The row marked zero CMBs represents those cases that had zero CMBs at time point one and developed a new CMB at time point two. * One female subject aged 46 years had a bleed in the first scan, but it had disappeared by the second scan. This accounts for the one subject with 0 CMB during scan 2.

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
