# Peer review of "Longitudinal Magnetic Resonance Imaging of Cerebral Microbleeds in Multiple Sclerosis Patients"

_diagnostics, 2020, doi:10.3390/diagnostics10110942_

Round 1

Reviewer 1 Report

This is an interesting clinical study to investigate the longitudinal f/u of CMBs in MS patients. In this manuscript, authors demonstrated that the increased number of CMBs in the MCS group between time points may indicate that MS patients may be at a higher risk to develop more CMBs in the longitudinal cohort. CMBs may serve as a biomarker for neurovascular disease in MS.

In spite of these interesting findings, some careful considerations should be made.

  1. In this article, do the authors want to emphasize that CMBs increase significantly in MS than HC, or do the authors want to highlight the advantages of the QSM method?

I am confused what the exact aim of this study is.

According to the previous article (reference 24; Radiology 2016, 281 (3), 884-895), your colleague has already announced that CMB increases in MS compared to HC.

In this regard, the purpose of this study is thought to be a story about the value and strength of the QSM method added to the existing study.

Then, the focus of the paper should report how much detection rate increased compared to when QSM was not used.

  1. Why does the author deal with both cohort 1 and cohort 2 in this paper? The two cohorts have different characteristics and imaging modalities. Especially, cohort 2 used a lower resolution.

  1. Why did the author evaluate longitudinal changes in CMB in MS?

In the discussion, more in-depth consideration is needed, not a simple description.

Because CMB is a biomarker for an aging process, it can be observed in all aging people. Then why CMBs are more frequent in MS patients than Healthy control?

Do CMBs appear differently depending on the stable or RRMS?

  1. CMBs located in the deep area (hypertensive) and CMBs located in the lobar area (Cerebral amyloid angiopathy) have different characteristics. Did the author distinguish it?

In table 2, among a total of 29 microbleeds, deep microbleeds are only 3.

As the authors noted in the limitation, CVD history shows a very important relationship with CMBs.

  1. In discussion, chapter 4.1 is too redundant.

  1. In conclusion, the suggestion that MS is a risk factor for CMB is too much to be skipped.

Since several confounding factors play a role in the occurrence of CMB, it is not appropriate to suggest like that.

Minor

  1. The layout of Figure 1 and Table 3 is bad for reading. Rearrange them so they can be seen within one page.

Author Response

We thank the reviewer for their comments. Please see the attached file for our responses.

Reviewer 2 Report

Thank you for asking me to review this manuscript. In a longitudinal study using susceptibility weighted imaging and mapping MRI, including a comparison/combination to other methods, they show that patients with MS might be at greater risk of cerebral micro bleeds. The manuscript is well written and scientifically seems very sound. Furthermore it is of potential clinical relevance. It would be nice to elaborate on why patients with MS might be at increased risk of CMBs and add a small section on the same. Otherwise I recommend the study for publication.

Author Response

We thank the reviewer for their comments. Please see the attached file for our response.

Response to Reviewer 2 Comments

We thank the reviewer for their comments. Our responses are provided below.

Point 1: It would be nice to elaborate on why patients with MS might be at increased risk of CMBs and add a small section on the same.

Response 1: Thank you for this comment. There has long been thought to be some thrombotic involvement in the veins in MS patients. Although clearly not true for all cases, this could be a manifestation of a breakdown of the BBB due to vascular permeability. We also assume that CMBs are caused by the consequence of leakage from blood vessels with disrupted vascular architecture. Also, the risk of developing CMBs is increased by decline in vascular health. This discussion point has been added to section 4.2.  

Round 2

Reviewer 1 Report

Title: Longitudinal Magnetic Resonance Imaging of

Cerebral Microbleeds in Multiple Sclerosis Patients

Manuscript ID: diagnostics-958331

I think it is meaningful to perform CMBs follow up longitudinally in MS patients. 

However, From the physician's point of view, this thesis focused too much on radiologic modality, and it is difficult to find a deep consideration on the meaning of CBS in MS.

  1. It is necessary to consider the meaning of CMBs in MS patients more in detail.

The author's answer did not cover my question sufficiently.

Do CMBs appear differently depending on the stable or RRMS? (previous review:Question 3)

  1. In this paper, is the imaging modality so important to be included in Table 1?

The clinical baseline characteristics of the patients included are considered to be much more important.

Round 3

Reviewer 1 Report

The author has adequately answered my question.